# MtDNA Analysis Indicates Human-Induced Temporal Changes of Serbian Honey Bees Diversity

**DOI:** 10.3390/insects12090767

**Published:** 2021-08-27

**Authors:** Marija Tanasković, Pavle Erić, Aleksandra Patenković, Katarina Erić, Milica Mihajlović, Vanja Tanasić, Ljubiša Stanisavljević, Slobodan Davidović

**Affiliations:** 1Department of Genetics of Populations and Ecogenotoxicology, Institute for Biological Research “Siniša Stanković”—National Institute of the Republic of Serbia, University of Belgrade, Bulevar Despota Stefana 142, 11060 Belgrade, Serbia; pavle.eric@ibiss.bg.ac.rs (P.E.); aleksandra@ibiss.bg.ac.rs (A.P.); katarina.eric@ibiss.bg.ac.rs (K.E.); slobodan.davidovic@ibiss.bg.ac.rs (S.D.); 2Center for Forensic and Applied Molecular Genetics, Faculty of Biology, University of Belgrade, Studentski trg 16, 11000 Belgrade, Serbia; milica.mihajlovic@bio.bg.ac.rs (M.M.); vanja.tanasic@bio.bg.ac.rs (V.T.); 3Center for Bee Research, Faculty of Biology, University of Belgrade, Studentski trg 16, 11000 Belgrade, Serbia; ljstanis@bio.bg.ac.rs

**Keywords:** honey bee, mtDNA, population genetics, haplotype diversity, subspecies

## Abstract

**Simple Summary:**

The western honey bee is one of the most economically and ecologically important species currently facing serious challenges in its whole area of distribution. The honey bee is a highly diverse species with about 30 subspecies that are adapted to regional climate factors, vegetation, pests and pathogens. The local populations of honey bees are rapidly changing and their diversity is constantly manipulated by beekeepers through the import of foreign queens, selection and migratory beekeeping. This manipulation may lead to such changes that honey bees lose their ability to thrive in the areas that were previously suitable for their wellbeing. To see how this human interference changed the genetic variability of native honey bee populations from Serbia, we sequenced part of the mitochondrial genome and compared them with published sequences. Our results suggest that human influence significantly changes the natural composition of honey bees in Serbia and that the presence of some previously reported subspecies could not be confirmed.

**Abstract:**

Local populations of *Apis mellifera* are rapidly changing by modern beekeeping through the introduction of nonnative queens, selection and migratory beekeeping. To assess the genetic diversity of contemporary managed honey bees in Serbia, we sequenced mitochondrial *tRNA^leu^-cox2* intergenic region of 241 worker bees from 46 apiaries at eight localities. Nine haplotypes were observed in our samples, with C2d being the most common and widespread. To evaluate genetic diversity patterns, we compared our data with 1696 sequences from the NCBI GenBank from neighbouring countries and Serbia. All 32 detected haplotypes belonged to the Southeast Europe lineage C, with two newly described haplotypes from our sample. The most frequent haplotype was C2d, followed by C2c and C1a. To distinguish *A. m. carnica* from *A. m. macedonica*, both previously reported in Serbia, PCR-RFLP analysis on the *COI* gene segment of mtDNA was used, and the result showed only the presence of *A.m. carnica* subspecies. An MDS plot constructed on pairwise *F_ST_* values showed significant geographical stratification. Our samples are grouped together, but distant from the Serbian dataset from the GenBank. This, with the absence of *A. m. macedonica* subspecies from its historic range of distribution in southern Serbia, indicates that honey bee populations are changing rapidly due to the anthropogenic influence.

## 1. Introduction

The western honey bee (*Apis mellifera* Linnaeus, 1758) is considered to be the most important managed pollinator species [1] Although native to the Old World (Africa, Asia and Europe) due to human activity, it is now spread through all continents except Antarctica [2]. *Apis mellifera* is a highly polymorphic species with approximately 30 described subspecies [3] of which ten are native to Europe [4]. Subspecies are often referred to as geographic races due to their distribution in specific geographic areas [2]. Additionally, several ecotypes and breeding lines exist within subspecies, reflecting significant variation and adaptation to regional climate factors, vegetation, pests and pathogens [5,6]. Based on morphological, genetic and ecological traits, all subspecies are divided into five evolutionary lineages of which Europe harbors four: the African lineage (A), the Southeast European lineage (C), the West and North European lineage (M) and the Near and Middle Eastern lineage (O) [7]. Honey bees, although long time human companions, cannot be considered as fully domesticated species since their mating system is not under complete beekeepers’ control [8], and queens mate in several flights with multiple drones in drone congregation areas [9,10]. Despite this still preserved mating autonomy, managed and feral honey bees colonies are under a strong anthropogenic influence [11]. Commercial breeding, along with the intensification of migratory beekeeping and queen importation, and as well as selection for desired traits, considerably influence the natural diversity and distribution range of honey bees [4,11]. This human interference usually results in hybridization and population admixture [12,13,14,15], but can also lead to depletion of genetic variability and local adaptations [4]. Changes in genetic diversity caused by the human introduction of nonnative subspecies, breeding lines and ecotypes are further propelled because of preserved mating autonomy and a single nonnative colony can be a potential source of hybrid individuals in colonies that are up to 15 km distant [16]. With intensive beekeeping, a high density of beehives per km^2^ and a small number of feral colonies, European honey bee populations are particularly prone to hybridization and population admixture [15,17,18] resulting in a considerable degree of genetic admixture among subspecies [19,20,21].

Not long after Rutner’s comprehensive morphological classification of *A. mellifera* subspecies [2], molecular tools have been developed to describe genetic diversity and disentangle the evolutionary history and classification of this species [22,23]. Since then, numerous nuclear and mtDNA markers were used for this task (reviewed in [6]) and sequencing of mitochondrial DNA region which contains a transfer RNA gene for leucine (*tRNA^leu^*), a noncoding insert and a partial sequence of the cytochrome oxidase unit II gene (*cox2*) is one of the most used methods in the literature [6,21,24,25]. Variation in the size of this region arose from the presence, absence or repetition of P and Q units, and is used to distinguish African (A) and European (C and M) lineages: in A lineage Po or P_1_ units are present, in M lineage there is only P unit and C lineage is notable for lack of any P unit. Distinguishing between A and O lineages is based on the presence of insertions, deletions and the number of Q elements [6,26]. Mitochondrial *tRNA^leu^-cox2* intergenic region sequences from different regions and subspecies are available on the NCBI GenBank and are an important resource that has helped improve the knowledge of native honey bee populations. However, due to human-induced rapid changes in the demographic history of western honey bees, new sequences are one of the best tools for the assessment of the human impact on the genetic variability of this species.

The genetic diversity of ten native European honey bee subspecies are the focus of numerous studies and research [27,28,29,30,31,32]. Although subspecies of A and O lineage are historically present in the Mediterranean area [3], the major body of *A. mellifera* subspecies belong to M in West and North Europe and C lineage in Southeastern Europe [3,7]. Despite a centuries-long tradition of beekeeping and constant human interference, subspecies’ specific genetic footprints are still preserved in some parts of Europe [4,33] but they are in a constant threat of hybridization, introgression and admixture which are detected in almost all analyzed populations [13,15,24,33,34,35]. The genetic diversity of honey bee populations in central and southeastern Europe, namely the Balkan Peninsula and surrounding countries, which is a natural area of distribution of four widely distributed subspecies *A. m. ligustica*, *A. m. carnica*, *A. m. cecropia* and *A. m. macedonica*, is thoroughly investigated in some parts but lacking in others [12,31,32,33,36,37,38] In addition, due to the constant human interference, modernization and changes in beekeeping practices, the genetic structure of honey bee populations is rapidly changing.

Serbia, geographically in the middle of the natural area of distribution of the South Eastern European lineage C, has a long tradition of beekeeping and strong breeding and beekeeper organizations. Serbian honey bees are thoroughly characterized on morphological, genetic, etiological and ecological levels, and based on this analysis two subspecies and three ecotypes were described. *A. m. carnica* was most prevalent in the northwest part of the country, *A. m. macedonica* in the southeast [33,36,39] and their hybrids were present both in the intermediate zone and in areas of two subspecies [12,33,36]. Based on chromosomal polymorphisms, three ecotypes of *A. m. carnica* (Banatski, Sjeničko-Pešterski, and Timočki) were described [40] and locally adapted populations were noted. Both the mtDNA and microsatellite variability of Serbian honey bees are well described. A highly variable mtDNA *tRNA^leu^-cox2* intergenic region was used in several studies [12,36,39,41] and more than seven haplotypes were reported showing significant genetic variation and possible ways of introduction of different mitotypes. Analysis of microsatellite variability [33,42] showed clinal south–north and west–east distribution of two distinct honey bee populations that corresponds to *A. m. carnica* and *A. m. macedonica* and their hybrids. Furthermore, in [33] authors showed that individuals from the north part of Serbia belong to a distinct genetic cluster characterized as *carnica-2 ecotype* and those from the south belong to a different distinct genetic cluster characterized as *macedonica-1 ecotype*.

However, all molecular investigations on both microsatellite and mtDNA loci, even the most recently published [33,42], are based on samples from the first decade of the 21st century. Since then, beekeeping practice in Serbia has significantly changed and the number of beekeepers vastly increased partially due to the strong government support of beekeeping. The latest official data from the Association of Beekeeping Organizations of Serbia, the leading beekeeping organization in the country, states the presence of 1,295,545 registered beehives and 25,830 registered beekeepers in 2018. According to our field data, the traditional way of beekeeping in Serbia is lost, and all beekeepers implement modern techniques in managing their colonies. The number of queen breeding institutions focused on desired traits of *A. m. carnica* is growing and Serbian legislation specifically allows breeding of this subspecies only (Law on animal breeding, 2009.) [43], in order to preserve its autochthonous status. Long-distance migratory beekeeping became more prevalent, especially migration from the south to the agricultural north part of the country, during flowering seasons of important agricultural plants. Additionally, with the rise of the number of beekeepers, the importation of Carniolan queens from Slovenia intensified in the past decade.

To shed a light on the current status of genetic diversity and maternal lineages of Serbian managed honey bees we conducted extensive analysis of mtDNA *tRNA^leu^-cox2* intergenic region of individuals sampled from apiaries from northern and southern parts of Serbia and compared them with existing GenBank deposits from surrounding countries and previously deposited from Serbia.

## 2. Materials and Methods

### 2.1. Sampling

For this study, 8 localities from southern and northern parts of Serbia were chosen, four in the South (Leskovac (L), Vlasina (V), Stara Planina (SP), Tromeđa (T)) and four in North (Subotica (S), Vršac (Vr), Deliblatska peščara (DS) and Fruška gora (FG)) (Figure 1 and Appendix A). All localities except T, which is located in between SP, L and V, are distinct geographical landmarks in Serbia. A total of 241 worker bees from 46 stationary apiaries were collected during late August and early September in 2020 and stored in 95% ethanol at −20 °C for further analysis. Apiaries were chosen according to the following criteria: they needed to be at least 15 years old and preferably inherited from the older family member or beekeeper, they needed to be stationary and with minimal or no queen importation. Approximately five worker bees from the apiary were chosen for genetic analysis, each representing one beehive.

A total of 1696 sequences from neighbouring countries and Serbia originating from different time periods and deposited in NCBI GenBank [12,24,33,36,39,41,44,45,46,47,48,49] were used for comparison (Appendix A).

### 2.2. DNA Extraction and PCR-RFLP Analysis

Whole-genomic DNA was extracted according to the protocol described in [25] and its concentration and quality were checked both with a spectrophotometer (NanoPhotometer, IMPLEN, Germany) and agarose gel electrophoresis.

To distinguish *A. m. carnica* from *A. m. macedonica*, the PCR-RFLP method described in [32] was used. Primers 5′GATTACTTCCTCCCTCATTA3′ [50] and 5′AATCTGGATAGTCTGAATAA3′ [39] were used for amplification of this mtDNA *COI* segment. Program for the amplification of the *COI* region for RFLP analysis consisted of the following steps: initial denaturation at 94 °C for 5 min, followed by 35 cycles of 45 s at 94 °C, 35 s long annealing step at the temperature of 57 °C and 60 s elongation at 72 °C. The step of final elongation was performed at 72 °C for 10 min. After amplification, the PCR products were purified according to [51], digested with restriction enzymes *Nco*I and *Sty*I and then visualized on 2% agarose gel stained with ethidium bromide under UV light. Expected restriction patterns for *A. m. macedonica* when digested with *Nco*I is 595 bp + 434 bp and when digested with *Sty*I is 626 bp + 403 bp, as described in [32]. For *A. m. carnica*, both *Nco*I and *Sty*I have no specific restriction site, and the whole 1029 bp fragment remains uncut.

### 2.3. tRNA^leu^-cox2 Intergenic Region Sequencing

The method used to identify the maternal origin of the honey bee colonies and to allocate them to different evolutionary lineages was based on the variation in the mitochondrial intergenic region located between the *tRNA^leu^* and *cox2* genes [22]. Region of interest was amplified with E2 (5′GGCAGAATAAGTGCATTG3′) and H2 (5′CAATATCATTGATGACC3′) primer pair [22] using a program described in [52]. Sequencing was performed using the same primers as for the PCR amplification. The Sanger sequencing of the amplified products was carried out using BigDye Terminator v. 3.1 sequencing kit (Applied Biosystems, Foster City, CA, USA) on ABI 3130 Genetic Analyzer (Applied Biosystems, Foster City, CA, USA) in the Center for Forensic and Applied Molecular Genetics, Faculty of Biology, University of Belgrade.

### 2.4. Statistical Analyses

All sequences used in analyses were aligned using MEGA 10.0.4 software [53]. The standard parameters of genetic diversity (the number of haplotypes, the number of polymorphic sites, haplotype diversity, nucleotide diversity, random match probability (RMP) and the mean number of pairwise differences (MPD)) were calculated using Arlequin ver. 3.5.2.2 software [54]. The RMP parameter is used for expressing the probability that two randomly sampled individuals from a population have a matching genotype and is calculated as the sum of square frequencies [55]. The MPD is a parameter that represents the measure of differences between all pairs of haplotypes in the sample. The same software was used for assessing genetic differentiation among populations by the analysis of molecular variances (AMOVA) and estimating pairwise population and overall *F_ST_* values. The statistical significance of all performed tests was assessed with 10,000 permutations. Multi-dimensional scaling (nonmetric MDS) implemented in PAST 3.25 software was used for the visualisation of the matrix of pairwise population *F_ST_* values [56]. Frequencies of different mtDNA haplogroups found in analyzed populations were calculated and used for the PCA to assess the contribution of each haplogroup to the genetic differentiation of the populations. PCA was performed using PAST 3.25 software.

### 2.5. Phylogeography

Phylogeographic analysis was performed using 1937 mtDNA haplotypes found in 14 different *A. mellifera* populations (Appendix A). The phylogeographic network was constructed using the median-joining method and maximum parsimony calculations for post-processing available in software Network 10.2.0.0 (https://www.fluxus-engineering.com/network_terms.htm). The substitutions were weighted equally since no data of different evolutionary rates for the *tRNA^leu^-cox2* intergenic region for *A. mellifera* are available. Substitutions specific for each haplotype were determined as the difference in nucleotide positions in comparison to the reference mitogenome NC_001566 [57].

## 3. Results

### 3.1. PCR-RFLP Analysis

The size of all PCR amplified *COI* segments for digestion was 1029 bp. In our sample, digestion with both *Nco*I and *Sty*I showed no restriction pattern previously described as characteristic for *A. m. macedonica* specific mtDNA lineage. Since no restriction sites were observed after RFLP analysis, and we could not detect the presence of a specific haplogroup that was previously reported in southeast parts of Serbia, we presume that all individuals in our sample belong to *A. m. carnica* (Appendix A).

### 3.2. Genetic Diversity of Mitochondrial tRNA^leu^-cox2 Intergenic Region Sequences in Serbia

A total of nine different haplotypes belonging to the eastern Mediterranean C lineage were detected in our sample (Figure 2, Appendix A). Seven of the nine detected haplotypes were previously described and two from the northern part of the country (FG and S localities) are novel (Figure 2, Appendix A). The novel haplotypes were named according to current nomenclature as C2ce for sample 228_AP_SS and C2db for sample 282_AP_SS. The C2d haplotype is the most frequent and detected in all localities, followed by the C2e haplotype which is present in all but one locality. The third most common haplotype C1a was detected in three northern and three southern localities. The C2c haplotype was more common in the north and present in all northern and one southern locality. It is interesting to note that all three rare haplotypes (C2i, C2j and C2ac) detected in this study are present in SP locality. All sequences are deposited in GenBank, accession numbers: MZ780962-MZ781202.

The frequency of the most common C2d haplotype ranged from 0.42 in L to 0.86 in T, but when localities were grouped by the region its presence was equally distributed (Appendix A). C2e haplotype is more prevalent in the South and C2i in the North.

The greatest haplotype diversity was detected in Vr (0.6952) and lowest in FG (0.3034) (Appendix A). Nucleotide diversity ranged from 0.0007 in T to 0.0030 in S and the mean number of pairwise differences from 0.3809 in T to 1.6401 in S (Appendix A). The value of the mean number of pairwise differences detected in our sample was in the value range of this parameter detected in other populations analyzed in this study.

Results of the AMOVA when our samples were divided according to the localities are presented in Table 1.

Although AMOVA showed a very low value of genetic variance among localities (0.04), statistically significant 6.88% of the genetic variance can be attributed to this variance component. These differences are probably due to the presence of detected rare haplotypes in SP and novel in FG and S. When localities are grouped according to their geographical region, their differences are lost and the percentage of variation among localities within the region increases (Table 2).

L locality is the most divergent from others, with significant differences from two south and three north populations (Appendix A). Although a statistically significant pairwise *F_ST_* value was detected between regions its value is the lowest one in the sample suggesting a close relationship between regions (Appendix A).

These results indicate that as a whole, the Serbian managed honey bee population is pretty homogenous but those local populations maintain specific haplotypes and distinct haplotype frequencies that may be attributed to the close relationships between beekeepers and their frequent queen exchange as well as local-specific genetic variability.

### 3.3. Genetic Diversity of Mitochondrial tRNA^leu^-cox2 Intergenic Region Sequences from GeneBank and Comparison with Novel Serbian Sequences

The haplotype network of all 1937 analyzed sequences is shown in Figure 3.

The overall distribution of haplotype groups of 14 countries is presented in Appendix A. Haplotype C1a, previously described as characteristic for *A. m. ligustica*, has a frequency of almost 95% in Italy, the country of origin of this subspecies. However, in lower frequencies, it is present in Romania, Austria, Hungary, Croatia, Slovenia and Serbia. The C2c haplotype is the only haplotype detected in Ukraine, has very high frequencies in Slovenia and Austria and it is present in lower frequencies in Romania, Hungary, Croatia, Italy and Serbia. The C2d haplotype, previously described as characteristic for *A. m. macedonica*, is the only haplotype detected in North Macedonia and it is prevalent in Greece, Bosnia and Herzegovina, Montenegro, Serbia, Romania, Bulgaria and Hungary. The C2e haplotype is the most frequent in Croatia and is present in more than 10% in Montenegro and Hungary.

When haplotype frequencies from our dataset are compared to previously published ones from Serbia, the most notable difference is a decrease in the frequency of C2d haplotype and an increase in frequencies of other haplotypes (C2c, C2e and C1a).

Results of the AMOVA analysis of all sequences analyzed in this work (GenBank and our samples) are shown in Table 3. This result reflects the high genetic variability of *A. mellifera* populations in the Balkan Peninsula and surrounding countries.

Results of AMOVA analysis when our samples were compared to the available GenBank samples from Serbia are shown in Table 4 and although the statistically significant difference between historic and contemporary sequences can not be observed, statistical significance among localities in datasets are present.

However, when historical data from each region were compared to its modern counterparts, statistically significant change could be observed with a high percentage of variation (12.48 and 5.79%) that contributes to these differences (Appendix A).

Differentiation between previously reported mtDNA variability and our dataset was further corroborated with the analysis of pairwise *F_ST_* differences (Appendix A) visualized using nonmetric MDS and shown in Figure 4. As can be observed from the picture, the MDS plot positioned our sample (South and North Serbia) in the lower-left quadrant while the GeneBank deposits (Serbia*) were positioned in the upper-left quadrant.

To investigate temporal changes in haplotype distribution of *A. mellifera* populations in Serbia, additional analyses were performed and the results are presented in Figure 5, Appendix A and Appendix A. When previously deposited GenBank sequences from Serbia were divided according to geographical region and the year of sampling and then compared to our data, differences in both haplotype frequencies, distribution and significantly different pairwise *F_ST_* values were observed, retaining the same general line of differentiation as in the first analysis. Furthermore, when previously deposited sequences were divided according to the region (North and South) for all previously published data, clear separation along the first coordinate from our sample was observed (Figure 5).

The PCA performed using the mtDNA haplotype frequencies represented with the first and second PC covering the total variability of 87.04% showed a similar distribution pattern for the populations in the plot (Figure 6) as in Figure 5. The PCA demonstrates that the main contributors for the positioning of the populations in the plot are the haplotypes belonging to the haplogroups C2d, C2c and C1a showing the geographical stratification of the analyzed populations.

Appendix A summarizes the correlations between the positioning of the populations in the MDS and PCA plots with the geographical centroids of the countries from which the analyzed populations originate. A high positive correlation (0.64) between the first coordinate and latitude for Figure 4 was detected while for Figure 5 a somewhat lower positive correlation between the first coordinate and longitude (0.22) and a higher positive correlation between the second coordinate and longitude (0.40) was observed. These results indicate that there is a positive correlation between the geographical position/region and haplotypes diversity and distribution.

## 4. Discussion

The genetic variability of managed honey bees is under the strong anthropogenic influence due to selection for desired traits, queen importation and migratory beekeeping. Previous work [12,33,36,39,40,41,42] showed that Serbia harbors substantial genetic diversity of this species reflected in the presence of two subspecies *A. m. carnica* and *A. m. macedonica*, their hybrids and three ecotypes. In this study, we tried to assess the existing genetic variability of managed honey bee colonies based on the mtDNA intergenic *COI-COII* region. We used PCR-RFLP analysis of this mtDNA segment to confirm the presence of *A. m. macedonica* subspecies and sequence of the *tRNA^leu^-cox2* intergenic region to determine present genetic variability for this molecular marker. Furthermore, we compared our samples with existing deposits from neighbouring countries from NCBI GenBank and available literature. To investigate temporal changes in mtDNA variability, we compared our samples with earlier Serbian honey bee deposits in NCBI GenBank.

The genetic diversity of the mtDNA *tRNA^leu^-cox2* intergenic region in Serbian honey bees is rather high with nine detected haplotypes, of which two are novel. Genetic diversity between localities is reflected both with significant *F_ST_* values for some pairs of localities and AMOVA results, suggesting that each locality has its characteristic pool of mtDNA lineages. Although this may reflect natural genetic diversity, it also may reflect close personal contact between local beekeepers who frequently trade queens between themselves [58]. Another result that supports local beekeepers’ relationships is the increased value of the percentage of variation between localities when they are grouped according to the region.

Results of PCR-RFLP analysis could not confirm the presence of specific *A. m. macedonica* mtDNA lineage due to the absence of a distinct restriction pattern previously described for this subspecies [32,39] suggesting that all sampled individuals belong to *A. m. carnica* mitochondrial lineage. Since the C2d haplotype is still dominant in Serbian apiaries, we are hesitant to claim that *A. m. macedonica* is now extinct from its previous natural range of distribution in Serbia [33] without detailed morphological confirmation. However, our results suggest that this could be the most likely scenario and that pure *A. m. macedonica* characterized with the presence of specific mtDNA lineage could not be detected using molecular methods. Further analyses of morphological and other molecular markers such as microsatellites and SNP is necessary to disentangle the presence of different *A. mellifera* subspecies or their hybrids in Serbia. Due to its gentle nature, high productivity and reduced swarming, beekeepers prefer *A. m. ligustica* and *A. m. carnica* which are both naturally and anthropogenically the most widespread C lineage subspecies. In some areas, they are even perceived as a threat to other subspecies, especially *A. m. mellifera* [13,15,18,59] but to the other C lineage subspecies also [60,61,62] which may be the case in the Serbian honey bee population since *A. m. macedonica* is not beekeepers’ favourite. Furthermore, these results are in line with information obtained from the field since all beekeepers we visited are adamant that they, in concordance with Serbian legislation, manage *A. m. carnica* only. Although *A. m. carnica* characterization by beekeepers is mainly morphological; it is reasonable to assume that beekeepers hold to this practice and this subspecies became predominant in apiaries. Stricter implementation of Serbian legislation that only *A. m. carnica* subspecies can be present in apiaries, which is strongly encouraged by a leading beekeeper organization as well, could also be one of the reasons behind the observed loss of *A. m. macedonica.*

Although AMOVA results did not show significant differences between the Southern and Northern geographical regions of Serbia, for comparison with published Serbian data, we decided to keep these groups separate not only because it reflects previously reported existence of *carnica-2 ecotype* and *macedonica-1 ecotype* [33] but also due to the low but significant *F_ST_* value that indicates that there are some differences between broader geographical regions. This separation, however, did not influence the results of the MDS and PCA and both analyses show the same pattern. Previously published haplotypes from the Serbian *A. mellifera* population are clustered separately from our sample. Furthermore, historical data are more closely clustered with Greek and North Macedonian honey bee populations that are natural areas of distribution of *A. m. macedonica* [39,63] and newly obtained sequences are closer to Bulgaria, Croatia and Romania, natural areas of distribution of *A. m. carnica.* Together with changes in haplotype frequencies, reflected in decreased frequency of C2d haplotype and increased frequencies of, in literature recognized as characteristic *A. m. carnica* haplotypes, C2c and C2e, results strongly indicate a temporal change in mtDNA diversity of Serbian honey bees. Previously published data include samples from 2006–2012 and in some cases, it is not stated from which specific geographic locations each sample originates. However, comparison of present data with previously published sequences grouped according to the year of collection and region from which they are collected showed that there is a slow shift from the southern cluster for the earliest data towards the western cluster for the more recent data. In addition, when samples for South Serbia from literature data were compared to our samples from South Serbia, all performed analyses demonstrated significant differentiation between them. This differentiation occurred in a relatively short time period of approximately ten years. The same, but the more notable pattern is present for comparisons of the samples from the north since these datasets are more separate on both MDS plots and have a higher percentage of between samples variation. Beekeeping is a dynamic process and according to our field data the traditional way of beekeeping in Serbia gradually disappeared over the past decade, which may be one of the reasons for observed changes in genetic variability. Human-induced temporal changes in honey bee subspecies and genetic diversity were previously reported and recognized [24,64,65,66]. Many of these changes were explained by the intentional introduction of bee subspecies preferred by beekeepers [15] and sometimes these introductions have unintentional consequences [24].

Due to the constant anthropogenic influence, the genetic variability of the native honey bee population is fluid, and many populations are under threat of losing their natural gene integrity through hybridization and introgression of foreign queens [5]. MtDNA variability, due to its maternal inheritance, is a particularly suitable model for the detection of human interference in honey bee populations since beekeepers manipulate exclusively with queens [33] and introgression of beekeepers’ favourite species can be monitored and tracked through the changes of haplotype frequencies over the years. Our results showed that in the Serbian honey bee population the shift towards *A. m. carnica* mitotypes occurred in the past decade. Once almost exclusive, the C2d haplotype is now less common overall. One of the conditions for including apiaries in our study was minimal or no queen importation, but rare is a beekeeper who does not, as they say, “refresh blood” from time to time, and that newly obtained queens are almost exclusively *A. m. carnica* subspecies which may be one of the reasons of decreased frequency of C2d haplotype. *A. m. carnica* is native subspecies in a large part of the Serbian territory and previous work showed that it readily hybridized with *A. m. macedonica* in the whole country [42]. The importation of Carniolan queens from Slovenia could explain the noticed increase in the frequency of the C2c haplotype that is characteristic of this region. Although all detected changes in the genetic structure of Serbian honey bees may be contributed to human interference and changes in beekeeping practices, locally present haplotypes and their frequencies indicate that local genetic variation is still present.

## 5. Conclusions

Clear differentiation of our sample from literature data and GenBank deposits for Serbian *A. mellifera* population along with the absence of PCR-RFLP specific *A. m. macedonica* pattern is a strong indication that changes in beekeeping practices, such as those that have occurred in Serbia, may alter the genetic composition of autochthonous honey bee populations. Only by constant monitoring of the honey bee population and timely implementation of the proper measures for their protection and conservation preventing further losses of genetic and biological diversity can be achieved.

## Figures and Tables

**Figure 1 insects-12-00767-f001:**
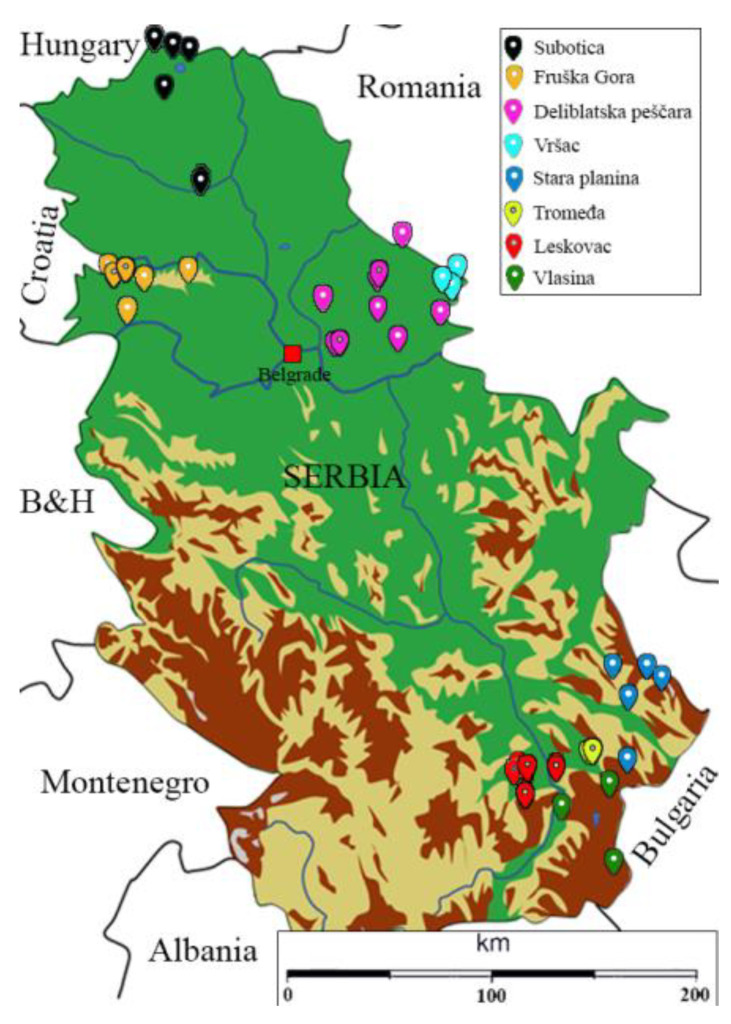
Sampling locations: Subotica (S), Fruška gora (FG), Deliblatska peščara (DS), Vršac (Vr), Stara Planina (SP), Tromeđa (T), Leskovac (L) and Vlasina (V).

**Figure 2 insects-12-00767-f002:**
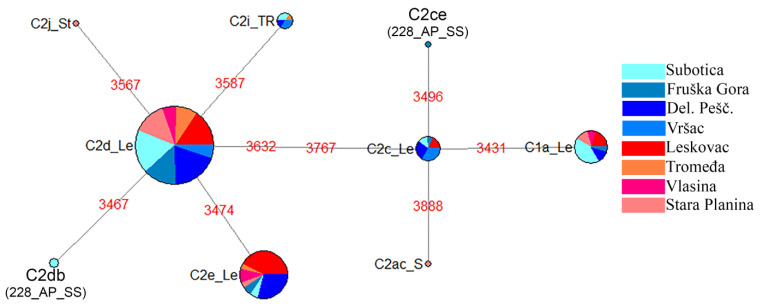
Median-Joining phylogeographic network of 241 mtDNA haplotypes in a Serbian *A. mellifera* sample based on the variability of the *tRNA^leu^-cox2* intergenic region (motifs in red). The size of the node is proportional to the number of individuals. The origin of each sample is shown in the legend.

**Figure 3 insects-12-00767-f003:**
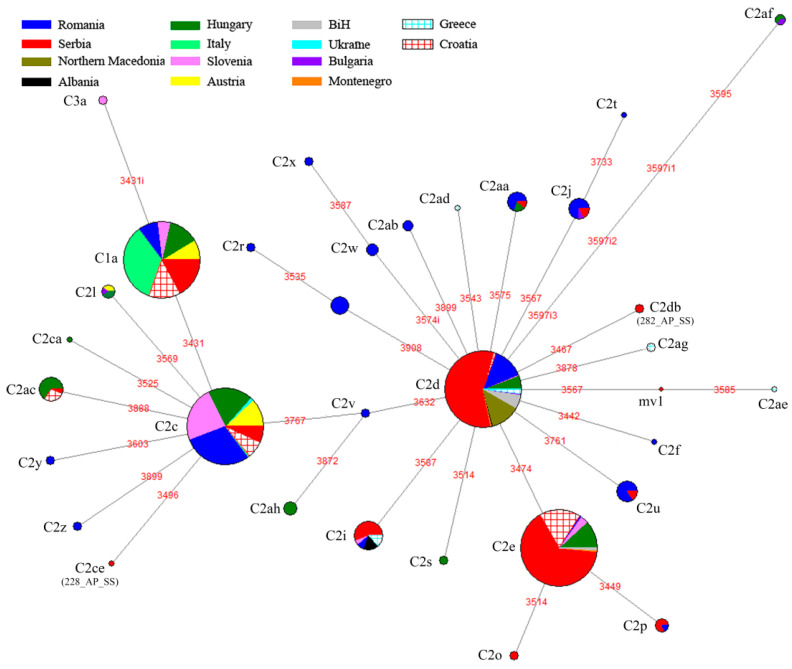
Median-joining phylogeographic network of 1937 mtDNA haplotypes detected in the 14 populations (Serbia- Romania, Austria, Hungary, Croatia, Slovenia, Bulgaria, B&H, Albania, North Macedonia, Montenegro, Greece, Ukraine and Italy) based on the variability of the *tRNA^leu^-cox2* intergenic region (motifs in red). Positions with insertions are marked with the suffix *i*. The size of the node is proportional to the number of individuals. The origin of each sample is shown in the legend. Serbia-population set represented by the literature and GenBank data for Serbia and 8 localities from our dataset (L, V, S, T, S, Vr, DS and FG).

**Figure 4 insects-12-00767-f004:**
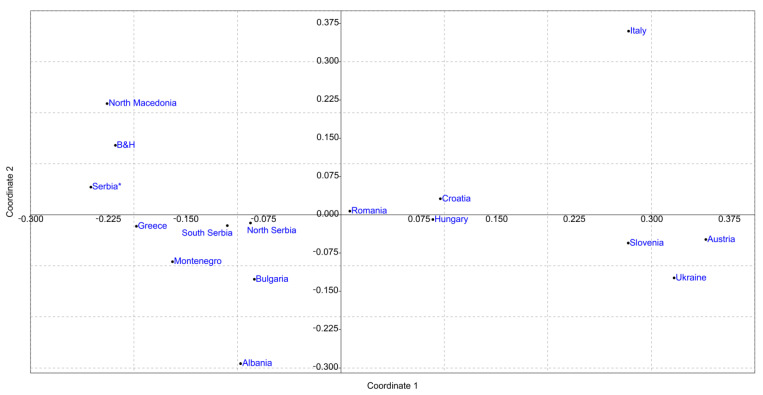
Nonmetric multidimensional scaling plot of *F_ST_* pairwise distances between the *A. mellifera* population of Serbia, two regions (North and South) and historical data (Serbia), and other *A. mellifera* populations based on the analysis of the variability of *tRNA^leu^-cox2* intergenic region. The goodness of fit is expressed with the stress value which is 0.1229 for this dataset. Population pairwise *F_ST_* values are presented in Appendix A.

**Figure 5 insects-12-00767-f005:**
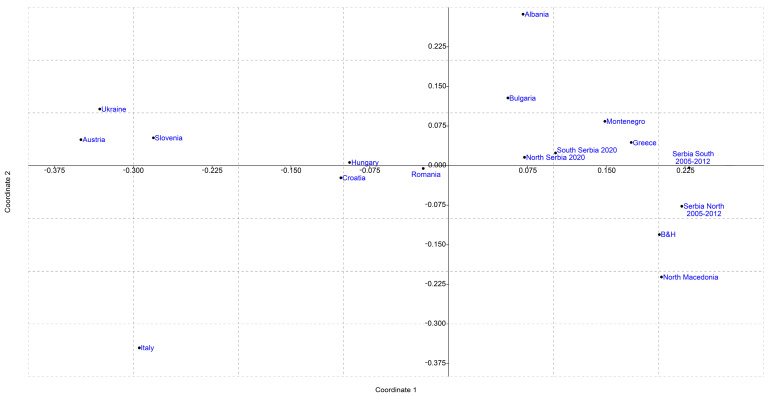
Nonmetric multidimensional scaling plot of *F_ST_* pairwise distances between the *A. mellifera* population of Serbia, two regions (North and South) and historical data divided according to the geographical region (North and South), and other *A. mellifera* populations based on the analysis of the variability of *tRNA^leu^-cox2* intergenic region. The goodness of fit is expressed with the stress value which is 0.1209 for this dataset. Population pairwise *F_ST_* values are presented in Appendix A.

**Figure 6 insects-12-00767-f006:**
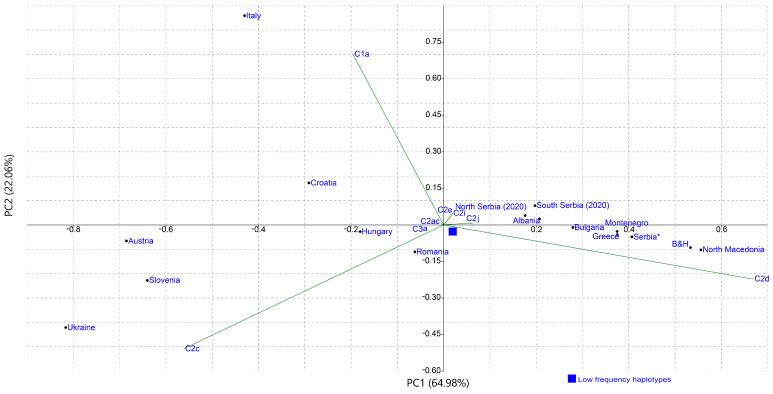
PC analysis based on mtDNA haplogroup frequencies in different *A. mellifera* populations. The contribution of each haplogroup to the PCs is shown with green lines. Frequencies of mtDNA haplogroups and references are listed in Appendix A.

**Table 1 insects-12-00767-t001:** Outcomes of AMOVA analysis based on the variability of the *tRNA^leu^-cox2* intergenic region analyzed for 8 localities in Serbia: L, V, SP, T, S, Vr, DS and FG.

Source of Variation	df	SS	Variance Components	Percentage of Variation
Among localities	7	12.223	0.04091	**6.88 (*p* = 0.00010)**
Within localities	233	129.038	0.554	**93.12 (*p* = 0.00000)**
Total	240	141.261	0.59472	-

df—degrees of freedom, SS—sum of squares, p—statistical significance; statistically significant results are in bold.

**Table 2 insects-12-00767-t002:** Outcomes of AMOVA analysis based on the variability of the *tRNA^leu^-cox2* intergenic region analyzed for 8 localities grouped in two regions South (L, V, S and T) and North (S, Vr, DS and FG) parts of Serbia.

Source of Variation	df	SS	Variance Components	Percentage of Variation
Among regions	1	1.727	−0.003	−0.56 (*p* = 0.475)
Among localities	6	10.469	0.043	**7.22 (*p* = 0.0002)**
Within localities	233	129.038	0.554	**93.34 (*p* = 0.0002)**
Total	240	141.261	0.593	-

df—degrees of freedom, SS—sum of squares, *p*—statistical significance, statistically significant results are in bold.

**Table 3 insects-12-00767-t003:** Outcomes of AMOVA analysis based on the variability of the *tRNA^leu^-cox2* intergenic region analyzed for 14 countries (Serbia*, Romania, Austria, Hungary, Croatia, Slovenia, Bulgaria, B&H, Albania, North Macedonia, Montenegro, Greece, Ukraine and Italy) and 8 localities from Serbia (L, V, S, T, S, Vr, DS and FG). Serbia*—population set represented by the literature and GenBank data for Serbia.

Source of Variation	df	SS	Variance Components	Percentage of Variation
Among countries	15	437.273	0.26342	**37.83 (*p* = 0.00000)**
Within countries	1921	831.479	0.43284	**62.17 (*p* = 0.00000)**
Total	1936	1268.752	0.69626	-

df—degrees of freedom, SS—sum of squares, *p*—statistical significance, statistically significant results are in bold.

**Table 4 insects-12-00767-t004:** Outcomes of AMOVA analysis based on the variability of the tRNAleu-cox2 intergenic region analyzed for Serbia* and a sample from Serbia presented in this paper. Serbia*—population set represented by the literature and GenBank data for Serbia.

Source of Variation	df	SS	Variance Components	Percentage of Variation
Among datasets	1	9.986	0.01912	5.58 (*p* = 0.33307)
Among localities in datasets	1	1.727	0.011189	**3.47 (*p* = 0.04584)**
Within localities	842	262.587	0.31186	98.95 (*p* = 0.00000)
Total	844	274.301	0.34288	-

df—degrees of freedom, SS- sum of squares, *p*—statistical significance, statistically significant results are in bold.

## Data Availability

Data are available on request from the corresponding author.

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
