# Peer review of "MtDNA Analysis Indicates Human-Induced Temporal Changes of Serbian Honey Bees Diversity"

_insects, 2021, doi:10.3390/insects12090767_

Round 1
Reviewer 1 Report
The paper entitled "MtDNA analysis indicates human induced temporal changes of Serbian honey bees diversity" aims to demonstrate temporal changes in mitochondrial diversity in the Serbian honey bee population. The article is well organized and presents an adequate literature review. However, there are several points that need to be addressed before publication:
- Mitochondrial DNA is maternally inherited, and both subspecies (carnica and macedonica) share mitochondrial haplotypes, so this technique should be combined with other techniques such as geometric morphometry or nuclear markers (microsatellites or SNPs), in order to better assess the existence or not of both subspecies. Because an asymmetric introgression may be occurring, being greater at the mitochondrial level, but the macedonica subspecies can still be detected with other markers.
- On the other hand, I believe that the authors should name the new haplotypes detected according to the most commonly used nomenclature (C2... or C3..., as appropriate).
- And the third important issue, is that the authors do not indicate from which localities and regions (north or south) are the Serbian samples downloaded from Genebank. I consider that the temporal analyses of the Serbian samples should be repeated only taking into account the same regions (north and south) to make the analyses comparable.
Other issues:
-Revise the scientific names, on line 72 it should read A. mellifera, and in the whole text of the article and in the supplementary material a space between A.m. is missing, it should read A. m.
- on line 77 the abbreviation for cytochrome oxidase II (cox2) should be included.
- The introduction is well written, but between lines 78-82 the composition of the intergenic region of the O lineage should also be included, and on line 130 the year of entry into force of the legislation should be included, to know how much time has passed since its implementation.
- In the material and methods, the year (2020) of sampling should be included, and in Table S2, in addition to the references of the works, the years of sampling should be included, in order to know how much time has elapsed.
- Figure 1 should include the abbreviations of the localities, better mark the borders, and indicate the north and scale.
- The results are well presented, although they should include the analyses mentioned above. In Figure 2, order the legend from north to south and include abbreviations, and in Tables 1, 2, 3 and 4 explain in the legend the significance levels and what the numbers in parentheses mean.
- In Figure 3, explain what the * in Serbia means.
- in Table 4, include the sample sizes, these analyses together with the MDS and PCA should be repeated taking into account only the same regions in the old samples.
- In the discussion, include these new analyses and references to the time elapsed between the implementation of the legislation and the years of the old and current sampling.
- In tables S3, S5 and S6 change commas to periods.
Author Response
The authors are very grateful for all valuable comments and suggestions that improved our work.
Comment1: Mitochondrial DNA is maternally inherited, and both subspecies (carnica and macedonica) share mitochondrial haplotypes, so this technique should be combined with other techniques such as geometric morphometry or nuclear markers (microsatellites or SNPs), in order to better assess the existence or not of both subspecies. Because an asymmetric introgression may be occurring, being greater at the mitochondrial level, but the macedonica subspecies can still be detected with other markers.
Author answer:
Thank You for this valuable comment. We absolutely agree that distinguishing between two subspecies should not rely on a single molecular marker and that other techniques should be implemented. We didn’t perform the morphometrical analysis which would be another way of confirming the existence of pure A. m. macedonica or its hybrids, but using the previously accepted method for the detection of pure A. m. macedonica mtDNA lineage we were unable to confirm its presence. Additional sentences were added in the Discussion section implementing Your suggestion. However, we must add that our work on microsatellite loci on the same sample we presented in this manuscript did not show north-south geographical stratification and grouping of our samples according to previously described carnica and macedonica subspecies clusters. Our microsatellite analysis rather showed population admixture suggesting that pure macedonica lineage is absent from our sample. Preliminary results can be found at https://sciforum.net/paper/view/10720.
Comment 2: On the other hand, I believe that the authors should name the new haplotypes detected according to the most commonly used nomenclature (C2... or C3..., as appropriate).
Author answer:
Thank You for this valuable comment. The novel haplotypes are named according to current nomenclature as C2ce for sample 228_AP_SS and C2db for sample 282_AP_SS and implemented in proper places in manuscript.
Comment 3: And the third important issue, is that the authors do not indicate from which localities and regions (north or south) are the Serbian samples downloaded from Genebank. I consider that the temporal analyses of the Serbian samples should be repeated only taking into account the same regions (north and south) to make the analyses comparable.
Author answer:
Thank You for this valuable comment that significantly improved our work. We added additional analysis taking into account previously published sequences from the north and south regions and implemented our findings in the Results and Discussion sections. Additional Supplementary Tables and Figures are added. This analysis further strengthens our claim that changes in genetic diversity of managed honey bees occurred in the past decade.
Other issues:
- Revise the scientific names, on line 72 it should read A. mellifera, and in the whole text of the article and in the supplementary material a space between A.m. is missing, it should read A. m.
We apologize for so many typographical errors, but it seems that some issue occurred between different Word versions. This has been corrected throughout the text.
- on line 77 the abbreviation for cytochrome oxidase II (cox2) should be included.
This has been updated.
- The introduction is well written, but between lines 78-82 the composition of the intergenic region of the O lineage should also be included,
Thank You for Your comment, an additional sentence improving the introduction has been added.
- on line 130 the year of entry into force of the legislation should be included, to know how much time has passed since its implementation.
Thank You for Your comment, year and reference for the Low on animal breeding are added to the text.
- In the material and methods, the year (2020) of sampling should be included, and in Table S2, in addition to the references of the works, the years of sampling should be included, in orde to know how much time has elapsed.
Thank You for Your comment, this has been updated. Table S2 now has two new columns: Year of sampling and Region, to see from which part of the country in each specific study, was the sample.
- Figure 1 should include the abbreviations of the localities, better mark the borders, and indicate the north and scale.
Thank You for this suggestion, Figure 1 has been updated.
- The results are well presented, although they should include the analyses mentioned above. In Figure 2, order the legend from north to south and include abbreviations, and in Tables 1, 2, 3 and 4 explain in the legend the significance levels and what the numbers in parentheses mean.
Thank You for Your comment. Results of additional analysis are added to the Result section and in corresponding Supplementary Tables and Figures. Figures and Tables have been updated.
- In Figure 3, explain what the * in Serbia means.
This has been updated.
- in Table 4, include the sample sizes,
This has been updated.
- these analyses together with the MDS and PCA should be repeated taking into account only the same regions in the old samples.
These analyses have been repeated and updated, and new results are presented in Results and Supplementary material.
- In the discussion, include these new analyses and references to the time elapsed between the implementation of the legislation and the years of the old and current sampling.
Thank you for this comment that greatly improved our manuscript. Additional sentences improving the discussion have been added, including a discussion of new analyses and references according to the sampling time and the time elapsed between the implementation of the legislation.
- In tables S3, S5 and S6 change commas to periods.
We corrected typographical errors according to Reviewer suggestions.
Reviewer 2 Report
The submitted paper is very interesting and the research on mtDNA was very well conducted. However, I don't understand why the Authors concluded and reported in all manuscript why "human-induced temporal changes of Serbian honey bees diversity". This is no immediately highlighted by the article results. There are other factors involved?
Besides, English and text must be revised. There are some grammatical errors and typos.
Author Response
The authors are very grateful for Your valuable comments and suggestions that improved our work.
Comment 1: The submitted paper is very interesting and the research on mtDNA was very well conducted. However, I don't understand why the Authors concluded and reported in all manuscript why "human-induced temporal changes of Serbian honey bees diversity". This is no immediately highlighted by the article results. There are other factors involved?
Author answer:
Thank you for Your comment. Additional analysis were performed and the Results and the Discussion sections now include additional explanations of observed temporal changes in managed Serbian A. mellifera populations which we believe have occurred due to the changes in beekeeping practices and stricter implementation of current Serbian legislation.
Comment 2: English and text must be revised. There are some grammatical errors and typos.
Author answer:
We apologize for so many typographical errors, but it seems that some issue occurred between different Word versions. We corrected typographical errors and revised the text.
Reviewer 3 Report
In the presented manuscript, Tanasković et al. examined the species diversity of Apis mellifera over time in Serbia by sequencing a portion of the mitochondrial genome of a number of samples and comparing this to previous sequenced data. The paper is relatively simple, straight forward and interesting.
I have few major issues with the paper & some minor comments, otherwise I think the manuscript is close to sufficient for publication.
Primarily, my main issue with the study is the use of only one loci in the mitochondria, given the mating system of honey bees. I would be interested to see how diversity has changed over time & geographically on the autosomes, and if the autosomes differ from the mitochondria. I realise this may not be possible, given the limited availability of autosomal markers from the previous samples, but it would definitely augment the results.
Additionally, the manuscript title & summary suggests a more elaborate examination of changes in diversity over time than is actually performed. The authors should make the temporal differences more obvious in the manuscript (such as stating the year of collection for each sample). Additionally, an analysis comparing changes in diversity over time could be performed (diversity ~ year of collection + collection location).
Otherwise I have a few minor comments:
Line 20: “Serbiawe” should read “Serbia, we”
Line 34: Should be “An MDS plot”, also it may be helpful to state what this stands for, as it is the first time the initialism is used in the manuscript.
Line 52: What is pairwise diversity for Apis mellifera, and how does it compare to a human population or Drosophila population? May give readers come context on how diverse they are.
Figure 1: Maybe a cartoon map would be more use to readers, given that a majority of the locations on the map are unreadable & will likely be read printed smaller than is seen on a screen. With a cartoon map you could label the nearest city to each collection point for reference (such as Sarajevo, Pristina, Sofia & Timisoara) & also shade areas or use contour lines to highlight changes in elevation & geographic barriers.
Table S2: If possible, could you provide the NCBI project number for each study where these sequences were taken from? To simplify things for readers. If no single ID is available for each study, then this may not be possible or necessary.
Figure 4: How does geographic distance compare to Fst distance? I would be interested to see how these compare & if there’s a correlation. If you took into account haplotype as a covariate, you could also determine if different covariates are more or less correlated than others.
Author Response
The authors are very grateful for all valuable comments and suggestions that improved our work.
Comment 1: Primarily, my main issue with the study is the use of only one loci in the mitochondria, given the mating system of honey bees. I would be interested to see how diversity has changed over time & geographically on the autosomes, and if the autosomes differ from the mitochondria. I realise this may not be possible, given the limited availability of autosomal markers from the previous samples, but it would definitely augment the results.
Author answer:
Thank You for this valuable comment. We absolutely agree that strong and definite conclusions on the variability of Serbian A. mellifera populations should not rely on a single molecular marker and that other techniques should be implemented. We are trying to obtain autosomal markers from previous work in order to disentangle genetic structure of our populations, but we are still unable to gain access to the previous samples. However, we must add that our work on microsatellite loci on the same sample we presented in this manuscript did not show north-south geographical stratification and grouping of our samples according to previously described carnica and macedonica subspecies clusters. Our microsatellite analysis rather showed population admixture suggesting that pure macedonica lineage is absent from our sample. Preliminary results can be found at https://sciforum.net/paper/view/10720.
Comment 2: Additionally, the manuscript title & summary suggests a more elaborate examination of changes in diversity over time than is actually performed. The authors should make the temporal differences more obvious in the manuscript (such as stating the year of collection for each sample). Additionally, an analysis comparing changes in diversity over time could be performed (diversity ~ year of collection + collection location).
Author answer:
Thank You for this valuable comment that significantly improved our work. We added additional analysis taking into account previously published sequences from the north and south regions and year of collection and implemented our findings in the Results and Discussion sections. Additional Supplementary Tables and Figures are added and Table S2 updated with additional information about location and year of sampling for each cited paper. This analysis further strengthens our claim that changes in genetic diversity of managed honey bees occurred in the past decade.
Other minor comments::
- Line 20: “Serbiawe” should read “Serbia, we”
This has been corrected.
- Line 34: Should be “An MDS plot”, also it may be helpful to state what this stands for, as it is the first time the initialism is used in the manuscript.
Thank You for this suggestion, additional explanation was included in the Material and Method section.
- Line 52: What is pairwise diversity for Apis mellifera, and how does it compare to a human population or Drosophila population? May give readers come context on how diverse they are.
Pairwise diversity for Apis mellifera populations used in this study are listed in Supplementary Table 4b and additional comparison between our sample and previously published data included in the Result section.
- Figure 1: Maybe a cartoon map would be more use to readers, given that a majority of the locations on the map are unreadable & will likely be read printed smaller than is seen on a screen. With a cartoon map you could label the nearest city to each collection point for reference (such as Sarajevo, Pristina, Sofia & Timisoara) & also shade areas or use contour lines to highlight changes in elevation & geographic barriers.
Thank You for this suggestion. Figure1 has been updated.
- Table S2: If possible, could you provide the NCBI project number for each study where these sequences were taken from? To simplify things for readers. If no single ID is available for each study, then this may not be possible or necessary.
Unfortunately, we were unable to find NCBI project numbers for each study where these sequences are taken, but more elaborate details about each study are added in Supplementary Table S2
- Figure 4: How does geographic distance compare to Fst distance? I would be interested to see how these compare & if there’s a correlation. If you took into account haplotype as a covariate, you could also determine if different covariates are more or less correlated than others.
Thank You for your valuable comment. Additional analysis were performed and included in Result section and Supplementary Table S9, showing high positive correlation between latitude and longitude and Fst distances.
Round 2
Reviewer 2 Report
in the new version provided, the Authors replied to all my comments. I encourage the publication.
Author Response
Thank You for Your comment, we appreciate all Your previous suggestion that greatly improved our work.
We must add that language has been corrected through the manuscript.